# Estrogen Signaling and Its Potential as a Target for Therapy in Ovarian Cancer

**DOI:** 10.3390/cancers12061647

**Published:** 2020-06-22

**Authors:** Simon P. Langdon, C. Simon Herrington, Robert L. Hollis, Charlie Gourley

**Affiliations:** 1Cancer Research UK Edinburgh Centre and Edinburgh Pathology, MRC Institute of Genetics and Molecular Medicine, University of Edinburgh, Edinburgh EH4 2XU, UK; simon.herrington@ed.ac.uk; 2The Nicola Murray Centre for Ovarian Cancer Research, CRUK Edinburgh Centre, MRC Institute of Genetics and Molecular Medicine, University of Edinburgh, Edinburgh EH4 2XU, UK; Robb.Hollis@ed.ac.uk (R.L.H.); Charlie.Gourley@ed.ac.uk (C.G.)

**Keywords:** ovarian cancer, estrogen, estrogen receptor, GPER, tamoxifen, letrozole

## Abstract

The estrogen receptor (ER) has functionality in selected ovarian cancer subtypes and represents a potential target for therapy. The majority (>80%) of high grade serous, low grade serous and endometrioid carcinomas and many granulosa cell tumors express ER-alpha (ERα), and these tumor types have demonstrated responses to endocrine therapy (tamoxifen and aromatase inhibitors) in multiple clinical studies. Biomarkers of responses to these drugs are actively being sought to help identify responsive cancers. Evidence for both pro-proliferative and pro-migratory roles for ERα has been obtained in model systems. ER-beta (ERβ) is generally considered to have a tumor suppressor role in ovarian cancer cells, being associated with the repression of cell growth and invasion. The differential expression of the specific ERβ isoforms may determine functionality within ovarian cancer cells. The more recently identified G protein-coupled receptor (GPER1; GPR30) has been shown to mediate both tumor-suppressive and tumor-promoting action in ovarian cancer cells, suggesting a more complex role. This review will summarize recent findings in this field.

## 1. Introduction

Ovarian cancer continues to be the most lethal gynecological cancer in the Western world, largely due to its frequently late diagnosis; approximately 300,000 new cases and 185,000 deaths are reported worldwide each year [1]. Ovarian cancer encompasses a broad range of different cancer types and has traditionally been categorized into epithelial ovarian cancers (EOCs) (90% of cases), germ cell tumors (5%) and sex cord-stromal tumors (2–5%). While there is relatively little controversy surrounding the cells of origin for the germ cell and sex cord-stromal tumors, the origins of most epithelial ovarian cancer subtypes are less well defined, and indeed, the majority are now considered to be likely to originate from outside of the ovary [2]. High grade serous ovarian carcinoma (HGSOC) is the most frequent subtype and accounts for 70% of EOCs, with endometrioid (10%), clear cell (10%), mucinous (5%) and low grade serous ovarian carcinomas (LGSOC) (5%) comprising the majority of the rest [3]. Granulosa cell tumors (GCTs) are the largest group of malignant ovarian stromal tumors and derive from granulosa cells, which are the cells responsible for estrogen biosynthesis in the ovary [4].

Estrogen signaling is mediated by several estrogen receptor isoforms. Estrogen receptor alpha (ERα, ESR1) was first identified in the late 1950s [5], while estrogen receptor beta (ERβ; ESR2) was reported in 1996 [6]. Both act (predominantly but not exclusively) in the cell nucleus. This so-called genomic signaling is further augmented by the non-genomic GPER1 (GPR30, GPER), which is a membrane-bound G-protein-coupled receptor capable of mediating both rapid and transcriptional events in response to estrogen [7]. The general structures of these receptors and key variants are shown in Figure 1.

Estrogen receptor alpha (ERα, ESR1) is the major mediator of the estrogen response; however, its activity needs to be considered alongside that of the other estrogen receptors [8,9]. In addition to the full length 66 kDa receptor (ERα66), two other truncated isoforms, ERα46 and ERα36, have been described [10], and the former (but not the latter) has been observed within ovarian cancer cells [11]. This isoform can be an antagonist of full-length ERα66 [12]. Several mutated forms of ERα have also been identified within ovarian carcinomas (Figure 1), and these will be discussed below.

Estrogen receptor beta (ERβ, ESR2) is known to be expressed as five different isoforms, namely ERβ1–5 (Figure 1), generated through alternative splicing. Only the full length form, ERβ1, is capable of binding agonist or antagonist ligands [13]. While ERβ2, ERβ4 and ERβ5 are unable to bind ligands, do not form homodimers and have no innate activities of their own, they can heterodimerize with ERβ1 (and also ERα) and induce transcriptional activity in a ligand-dependent manner [13]. ERβ3 is specific to the testis and not found in ovaries [14]. The cellular localization of each isoform (nucleus vs. cytoplasm) also appears to be an important determinant of activity [15].

GPER1 (G protein-coupled estrogen receptor 1, GPR30, GPER) is a membrane-bound G-protein-coupled receptor that binds estrogen and activates multiple downstream signaling pathways. These include the stimulation of adenyl cyclase and an increase in cAMP, promoting intracellular calcium mobilization and the synthesis of phosphatidylinositol 3, 4, 5-triphosphate [7].

It is apparent that several of the major forms of ovarian cancer—namely HGSOC, LGSOC, endometrioid ovarian carcinoma and adult-type GCTs—show greater sensitivity to estrogen and can respond to strategies that either inhibit the production of estrogen (i.e., aromatase inhibitors) or that directly compete with its action at estrogen receptors (i.e., anti-estrogens such as tamoxifen or fulvestrant), which may have therapeutic value in selected groups of ovarian cancer patients. This review will focus on estrogen signaling and its impact on the risk, incidence, development, progression and therapy of differing forms of ovarian cancer with an emphasis on the responsive ovarian cancer subgroups.

## 2. Estrogen and the Risk and Incidence of Ovarian Cancer

It has long been recognized that the risk of developing ovarian cancer is strongly linked to a woman’s reproductive history. Hence, ovarian cancer is more frequent in women who had early menarche or late menopause, and the risk is greater for women who have had no or few children or had their children at a later age [16]. While there are a number of possible associations that might explain these observations (e.g., the cumulative number of ovulations [17,18]), these events are also linked to an increased exposure to estrogen, and hence, it has been speculated that ovarian cancer risk might be associated with estrogen action [16].

The clinical use of exogenous estrogens as hormone replacement therapy (HRT) provides direct evidence of a small but increased risk of developing ovarian cancer. A meta-analysis of 52 epidemiological studies investigating the long-term use of hormone replacement therapy (HRT) in post-menopausal women identified an increased relative risk (RR) of ovarian cancer of 1.20 (confidence interval (CI), 1.15–1.26; *p* < 0.0001) for ever-users vs. non-users in prospective studies [19]. In current and recent users, the risk was 1.43 (CI, 1.31–1.56; *p* < 0.0001). Importantly, this increased risk was associated with the serous (RR = 1.53) and endometrioid (RR = 1.42) carcinoma sub-types but not mucinous (RR = 0.93) or clear cell (RR = 0.75) groups [19].

In contrast to this enhanced risk with HRT, the administration of estrogen as a component of oral contraceptives in premenopausal women is associated with a markedly reduced risk of developing ovarian cancer [20]. However, despite the high levels of circulating estrogen, this is likely explained by the decreased production of ovarian estrogen and reduced total number of ovulations, which is recognized as a major determinant of ovarian cancer risk [18,20].

Dietary exposure to estrogens may also modify risk. A meta-analysis has investigated the administration of phytoestrogens (plant-derived estrogens) in the diet and has associated high levels of specific phytoestrogens, e.g., isoflavones, with a reduced risk (RR = 0.63) of developing ovarian cancer consistent with a protective effect [21]. Further studies would be valuable to adequately control for other potential causes. However, in support, the isoflavone genistein has been studied against many ovarian cancer cell lines in vitro, demonstrating an inhibitory effect, with effects likely mediated via ERβ [22].

The normal ovary contains high levels of ERβ, with expression found predominantly in granulosa cells, theca cells, the surface epithelium and the corpus luteum [23]. Multiple studies have demonstrated a reduction of ERβ expression in epithelial ovarian cancers relative to in the normal ovary, and this led to the view that it might have a tumor suppressor function (although the normal ovary may not be the best tissue comparator for epithelial ovarian cancer) [24,25]. Several polymorphisms in ESR2 have been associated with a small increased risk of ovarian cancer [26,27,28]. These include the polymorphism rs1271572, which has been associated with an increased risk, particularly in younger women [26]; rs1256030, associated with ovarian cancer in Caucasians; and rs1256031, associated with ovarian cancer in Japanese women [27].

A shift from ERα to ERβ signaling could also be a factor in the transformation from endometriosis to endometriosis associated ovarian cancer (EAOC). Although generally benign, endometriotic lesions can undergo malignant transformation. Studies are underway to investigate molecular changes in the progression of endometriosis to EAOC, and this has been suggested to involve modified estrogen signaling [29]. In endometriosis, estrogen regulation seems to occur through the activation of ERβ rather than ERα, and analysis of gene expression data suggests that ERα signaling becomes inactivated with the shift from endometriosis to EOAC [29].

## 3. Estrogen Receptor Expression in Epithelial Ovarian Cancer

### 3.1. ERα Expression in Ovarian Carcinomas

Multiple studies have investigated the expression of ERα in epithelial ovarian cancer, but the largest is the study reported by Sieh et al. in 2013 [30]. This investigated 2933 women and identified ERα positivity in 81% of HGSOCs, 88% of LGSOCs and 77% of endometrioid ovarian carcinomas [30]. By contrast, expression was detected in only 21% of mucinous carcinomas and 20% of clear cell carcinomas. This is in line with the increased risk of HRT being associated with serous and endometrioid carcinoma but not mucinous or clear cell carcinoma [19]. Both ERα and progesterone receptor (PR) expression were strongly associated with improved survival for endometrioid ovarian carcinoma, while PR (but not ERα) was associated with favorable outcomes in HGSOC [30]. The low expression of ERα in clear cell carcinomas has been ascribed to epigenetic repression [31].

The functionality of ERα in ovarian cancer cells has been demonstrated using ovarian cancer cell line models, wherein the requirement of a moderate to a high level of ERα expression for a growth response to 17β-estradiol (E_2_) and anti-estrogens has been shown both in vitro [32,33,34] and in vivo [35]. A direct comparison of the roles of ERα and ERβ, using isotype-selective agonists, has provided support for ERα mediating the growth response (and gene expression changes) when ERα is expressed at high levels [36]. This was confirmed in a separate study using both agonists and antagonists of ERα and ERβ, wherein the ERα agonist 4,4′,4″-(4-propyl-[1H]-pyrazole-1,3,5-triyl) trisphenol (PPT) stimulated growth while the ERα antagonist 1,3-bis(4-hydroxyphenyl)-4-methyl-5-[4-(2-piperidinylethoxy)phenol]-1H-pyrazole dihydrochloride (MPP) inhibited growth [37]. Moreover, the silencing of ERα through siRNA reduction neutralized the growth stimulation produced by added estrogen [37].

In addition to growth regulation, estrogen, via ERα, also promotes cell migration and a shift to epithelial–mesenchymal transition. These changes are mediated via the SNAIL- and SLUG-mediated down-regulation of E-cadherin [38]. Interestingly, this process is inhibited by ERβ [38]. The extracellular molecule fibulin, which binds to fibronectin and laminin, is also regulated by estrogen via ERα and has been proposed to have a key role in cell attachment and motility [39,40].

### 3.2. ERβ Expression in Ovarian Carcinomas

As mentioned above, the early evaluation of the role of ERβ suggested it to be that of a tumor suppressor, with expression associated with decreased proliferation and migration [23,24,25]. Further support is provided by ERβ-specific agonists, e.g., 2,3-bis(4-hydroxy-phenyl)-propionitrile (DPN), inhibiting the growth of ovarian carcinoma cells [37,41,42,43] and ERβ-specific antagonists, e.g., 4-[2-phenyl-5,7-bis(trifluoromethyl) pyrazolo [1,5-a]-pyrimidin-3-yl]phenol (PHTPP), enhancing growth [37]. In accord with this, the overexpression of ERβ results in reduced proliferation both in vitro [25] and in vivo [25] and repression of invasion and migration [44,45]. Conversely, the knockdown of ERβ with a specific siRNA increased cell growth [42]. Downstream signaling processes that have been associated with the ERβ-mediated inhibition of growth in ovarian cancer cells include the decreased expression of pRb [25], phospho-AKT [25], cyclin D1 [25] and cyclin A2 [25,44] and increased expression of p21 [44].

Further examination of the expression levels and cellular location of specific ERβ isoforms in ovarian cancers, together with functional studies using cell line models, has revealed some important fine detail [40,43,44,45]. Consistent with a tumor suppressor role, a higher expression of cytoplasmic ERβ1 (full length isoform) is significantly associated with better disease-free and overall survival [46]. The overexpression, through transfection, of ERβ1 in the SKOV3 ovarian cancer cell line [44] or in the ES-2 cell line [41] resulted in reduced proliferation and motility and increased apoptosis [44]. The overexpression in SKOV3 of variants of ERβ1 with deletions in the AF-1, DNA and ligand-binding domains had no effect on proliferation or motility, supporting the requirement for the full-length receptor [44]. The expression of these variants has been detected in ovarian cancers [44].

In contrast to the inhibitory role of ERβ1, both ERβ2 and ERβ5 have been associated with pro-migratory and invasive activities. HGSOC patients whose cancers had cytoplasmic ERβ2 had poorer survival, and this was linked to chemoresistance [47]. ERβ2 overexpression has been shown to increase cell migration and invasion but not proliferation in ovarian cancer cells [46]. The mitochondrial activity of cytoplasmic ERβ2 signaling in serous carcinomas has now been associated with binding to BAD, leading to reduced apoptosis, hence having a pro-survival role [48]. In comparison to serous cancers, cytoplasmic ERβ2 expression is reduced in clear cell carcinoma [46].

High levels of expression of nuclear ERβ5 are found in advanced ovarian cancers, notably in serous and clear cell carcinomas, and this is also associated with poor survival, although cytoplasmic ERβ5 is associated with better outcomes [46]. ERβ5 has been shown to enhance cell migration, invasion and proliferation [46].

In LGSOC, the prominent expression of nuclear ERβ2 and ERβ5 have been identified, with high levels of cytoplasmic ERβ2 in metastatic lesions [49]. The LGSOC cell lines, HOC-7 and VOA-1056, express ERβ1, ERβ2 and ERβ5; however, they did not demonstrate a growth response to 17β-estradiol, PPT or DPN (but ERα expression is low/negative), suggesting that LGSOC may have be less dependent on ERβ [49].

### 3.3. ERα/ERβ Expression Ratio in Ovarian Carcinomas

The ERα/ERβ expression ratio has been suggested to change in the progression from normal ovary to primary ovarian cancer to metastatic disease, with the loss of ERβ and an increase in ERα expression [25,50]. Since ERα and ERβ can mediate different signaling and functions, it is likely that the relative levels of expression of the two receptors is important in determining the final outcome. The decrease in the expression levels of ERβ1, ERβ2 and ERβ4 in ovarian cancers has been associated with the hypermethylation of the ERβ promoter. However, this is not the case for ERβ5 [51].

When both ERα and ERβ are co-expressed in tissues, they can form functional heterodimers [52,53,54]. The biological roles of ERα/β heterodimers in the presence of each respective homodimer are poorly described; however, ERβ has been shown to have a negative regulatory effect on ERα when forming heterodimers with ERβ in transfected breast cancer cell line models (MCF7 [52,53] and T47D [54]). When co-expressed, ERβ exhibits an inhibitory action on ERα-mediated gene expression and, in many instances, opposes the actions of ERα. In ovarian cancer cell line models that express both ERα and ERβ, the opposing effects of these two receptors has been demonstrated [37]. The SKOV3 and OV2008 cell lines express both receptor subtypes, and these were treated with subtype-specific modulators. Treatment with the ERα antagonist, MPP, or the ERβ agonist, DPN, suppressed the growth of both cell lines, while conversely, treatment with the ERα agonist, PPT, or the ERβ antagonist, PHTPP, increased cell growth [37]. The combined treatment with the ERα antagonist, MPP, and the ERβ agonist, DPN, had a synergistic effect in suppressing cell growth [37]. This study provides strong support for estrogen having a growth-promoting effect through ERα and growth-inhibitory effect via ERβ in ovarian cancer cells.

### 3.4. GPER1 Expression in Ovarian Carcinomas

GPER1 is widely distributed throughout normal tissues with high levels reported in the heart, liver, lung, intestines, brains and ovary [7]. The role of GPER1 in ovarian cancer is somewhat controversial, with differing studies demonstrating opposite outcomes. Ignatov and colleagues assigned GPER1 a tumor suppressor role in ovarian cancer [55]. Expression was reported to be lower in ovarian cancers than in benign or non-malignant tissue. Furthermore, expression was higher in early than in late stage cancers, and expression was associated with a more favorable clinical outcome. Consistent with a suppressor role, a selective GPER-1 agonist, G1, suppressed the proliferation of ovarian cancer cell lines (SKOV3 and OVCAR3) [55].

By contrast, Smith et al. demonstrated that GPER1 was associated with poor survival in 89 ovarian cancer patients, with expression more frequent in epithelial ovarian cancers than in tumors with low malignant potential [56]. A study by Zhu and colleagues of 110 patients with epithelial ovarian cancer suggested that nuclear, but not cytoplasmic, GPER1 expression was associated with poor overall survival [57]. In the OVCAR5 ovarian cancer cell line, both 17β-estradiol and G1 stimulated proliferation, with an increase in cells in the S phase and the up-regulation of c-fos and cyclin D1 [58]. Further studies using the SKOV3 ovarian cancer cell line indicated that GPER1 ligand-independently stimulated proliferation, migration and invasion [59].

A study by Albanito and co-workers suggested that interplay between ERα and GPER1 may also be important [60]. The use of ERα-positive BG-1 ovarian carcinoma cells showed that both 17β-estradiol and G1 (the GPER-1 agonist) induced c-fos expression and up-regulated cyclins D1, E and A. Both GPER1 and ERα were required for c-fos stimulation and ERK activation in response to either G1 or 17β-estradiol. The inhibition of the EGFR pathway inhibited c-fos and ERK activation, indicating that in ovarian cancer cells, the GPER1/EGFR signaling relies on ERα expression [60].

## 4. Estrogen Signaling in Granulosa Cell Tumors

Granulosa cell tumors (GCTs), although comprising <5% of ovarian cancers, are the largest group of malignant stromal tumors arising in the ovary. GCTs can be divided into adult-type GCTs, which constitute 95% of GCTs, and the relatively rare juvenile GCTs [4].

Adult GCTs are characterized by a mutation (C134W) in *FOXL2* detected in 92–97% of this subtype [61,62]. FOXL2 is a transcription factor that has a critical role in the proliferation, apoptosis and steroidogenesis of granulosa cells [61]. Normal granulosa cells synthesize 17β-estradiol via the use of aromatase (CYP19A1) [4]. The *FOXL2* C134W mutation results in CYP19A1 up-regulation, in turn leading to increased aromatization and excess estrogen production [63].

Evaluation of the roles of the ER subtypes in GCTs has suggested that ERβ expression may be of particular significance in these tumors. ERβ expression in normal granulosa cells is among the highest levels in different body tissues [64]. In GCTs, ERβ is expressed more strongly than ERα, which has led to the suggestion that ERβ is the dominant receptor [65]. The expression of ERβ1, ERβ2 and ERβ5 has been identified at moderate levels [65]. A role for mitochondrial ERβ2 acting as a binding partner of BAD in the apoptotic cascade, thereby inhibiting apoptosis, has also been described [65]. ERα expression is reportedly widespread in GCTs but at relatively low levels [66,67].

As for epithelial ovarian cancers, the role of GPER1 is poorly clarified, with different studies producing a mixed picture [68,69]. In a study reported by Heublein and colleagues, GPER1 protein expression was identified in 54% of GCTs, with a strong intensity of expression at primary diagnosis being associated with significantly reduced survival [68]. Francois and colleagues found expression in approximately 90% of GCTs; use of cell lines (KGN and COV434) derived from GCTs, indicated that 17β-estradiol did not affect cell growth but decreased the migration and invasion of GCT cells [69]. This was shown to be mediated via GPER1 and was linked to the inhibition of ERK1/2, which is frequently constitutively activated in GCTs [69].

## 5. Targeting ER with Anti-Estrogens and Aromatase Inhibitors

The use of hormonal therapy to treat epithelial ovarian cancer is now well established. The anti-estrogen tamoxifen (which targets the estrogen receptor) was initially used and continues to be an option; more recently, aromatase inhibitors (which act by inhibiting the conversion of androgen to estrogen, hence reducing the circulating levels of estrogen) have been trialed and shown to be effective. Investigations using in vitro and in vivo models of ovarian cancer have helped to consolidate the case for ER expression being critical to this response.

### 5.1. Cell Line Model Evidence for ERα as a Target for Therapy in Ovarian Cancer

Experimental studies using ovarian carcinoma cell lines have supported the idea that the level of expression of ERα is important as a determinant of the response to anti-estrogen therapy. The growth response of ovarian cancer cells to tamoxifen and pure anti-estrogens such as fulvestrant (faslodex; ICI 182,780) has been evaluated in ERα-high- and ERα-low-expressing ovarian cancer cell lines [32,33,34]. The estrogen (17β-estradiol) growth stimulus elicited in the ERα-high-expressing cells could be blocked by the addition of tamoxifen or fulvestrant, while tamoxifen and fulvestrant were ineffective against low-ERα-expressing cells [32,35]. Both drugs were also effective in vivo against an ERα-high ovarian cancer xenograft [35]. The silencing of ERα expression in the SKOV3 ovarian cancer cell line not only inhibited estrogen-stimulated growth but also reversed the inhibitory effects of 4-hydroxytamoxifen (the active metabolite of tamoxifen) and fulvestrant, consistent with ERα mediating the effects of these anti-estrogens [37]. A possible role for ERβ was tested by silencing ERβ in the cell line, but this had a minimal effect on the response to anti-estrogens [37].

The treatment of tumor explants from ERα-positive/ERβ-negative-HGSOC-patient-derived xenografts (PDXs) with either 4-hydroxytamoxifen or fulvestrant has also provided support for the idea of ERα mediating the effects of these drugs [70]. Of the four PDXS studied, the two xenografts with the higher levels of ERα expression demonstrated decreased proliferation (as measured by BrdUrd incorporation) upon treatment with drugs, while the two xenografts with lower ERα expression were unaffected, consistent with the requirement for a moderate–high level of ERα for a response [70]. Fulvestrant decreased ERα protein expression within the responding cells, in line with its mode of action [70].

While the above studies support a role for higher levels of ERα expression mediating the effects of anti-estrogens, the roles of ERβ and GPER1 remain relatively unexplored with respect to the response to tamoxifen and other anti-estrogens in this disease setting. It is clear from the studies described in Section 3.1, Section 3.2, Section 3.3 and Section 3.4 using specific agonists and antagonists of ERβ and GPER1 that these receptors can also mediate estrogen signaling; however, their response to the clinically used anti-estrogens has not been studied in ovarian cancer. Therefore, even if ERα has a dominant role in the growth response to clinical anti-estrogens, it is feasible that ERβ or GPER1 might demonstrate functionality in ERα-negative disease or alternatively modulate an ERα-mediated response. Further studies are required to assess this.

### 5.2. Overview of Clinical Trials Evaluating Tamoxifen and Aromatase Inhibitors in Ovarian Carcinoma

The selective estrogen receptor modulator tamoxifen has been used clinically to treat ovarian carcinoma since the early 1980s, and the overall mean response rate for this treatment is reported to be 10–15%, with a disease stabilization rate of 30–40% (reviewed in references [71,72,73,74]). Most of the patients in these clinical trials were heavily pretreated, and many studies did not select for ER positivity. As pointed out by Perez-Gracia and colleagues [72], the analysis of the use of tamoxifen in trials in which at least 50% of the patients had not received more than one prior treatment demonstrated an overall response rate of 26%, with a 9% complete response rate, which contrasted with clinical studies of heavily treated patients, where the response rate was only 4% [72].

Since 2002, the non-steroidal aromatase inhibitors, notably letrozole and anastrazole, have been investigated, and the level of antitumor activity appears to be comparable to that of tamoxifen. The most recent and comprehensive meta-analysis is the report by Paleari and colleagues, which analysed 53 endocrine therapy trials encompassing 2490 patients [74]. The clinical benefit rate (CBR; defined as the proportion of patients who showed a complete response, a partial response or stable disease) was a 41% CBR overall for all the endocrine therapies assessed, with a 43% CBR for tamoxifen (23 trials) and 39% CBR for aromatase inhibitors (10 trials), suggesting the equivalence of the two approaches [74].

A recent trial (PARAGON) evaluated the use of anastrazole in a phase 2 study of asymptomatic patients with ER-/PR-positive recurrent ovarian carcinoma with CA125 progression [75]. A response rate of 4% and clinical benefit rate of 35% were demonstrated, which are disappointing given that these patients had only low volume disease and had received only a single previous line of chemotherapy [75]. As pointed out in an associated editorial, the use of only a low ERα expression score for positivity and the mixture of different histological subtypes might have produced a sub-optimal outcome [76].

### 5.3. High Grade Serous Ovarian Carcinomas

Two recent reports have described studies evaluating the use of endocrine therapy in their respective centers and provide interesting insight into HGSOC outside of a trial setting [77,78]. An analysis of 97 patients treated at the Royal Marsden Hospital, London, investigated the use of tamoxifen and letrozole in high grade ovarian cancer (of which 91% were HGSOC) [77]. Despite more than a quarter of patients having had five or more lines of prior chemotherapy, with half of the patients having an unknown level of ER, there was a 60% clinical benefit rate (65% for tamoxifen and 56% for letrozole) [77]. The responders to letrozole had significantly longer duration of response.

In an analysis of 269 HGSOC patients studied within Edinburgh over a 25-year period, letrozole and tamoxifen had comparable overall responses (8% and 11%, respectively) and clinical benefit rates (41% and 33%, respectively) [78]. Patients with a high ER score (discussed further below) and a longer treatment-free interval were most likely to benefit [78]. The conclusion from both of these analyses is in line with the clinical trial results and is that treatment with either tamoxifen or letrozole is a rational treatment option for patients with ER-positive HGSOC, producing a comparable overall response rate, CBR and disease stability.

Heinzelmann-Schwarz and colleagues evaluated letrozole as maintenance treatment in a cohort of HGSOC patients [79]. The use of letrozole was associated with a significantly prolonged recurrence-free interval after 24 months of treatment (60% for letrozole (*n* = 23) vs. 39% for the control (*n* = 27); *p* = 0.035). This was also the case with patients being treated with letrozole alongside bevacizumab [79].

### 5.4. Low Grade Serous Ovarian Carcinoma

Low grade serous ovarian carcinoma (LGSOC) is poorly chemosensitive, so endocrine therapy may represent a promising alternative [80]. Gershenson and colleagues identified a 9% response rate and 61% disease stabilization rate in a retrospective analysis of 64 LGSOC patients who had received a total of 89 hormonal regimens [81]. This led to a further analysis of the outcomes associated with hormonal maintenance therapy compared with routine observation after cytoreductive surgery and platinum chemotherapy in women with stage II to IV LGSOC [82]. The progression-free survival for patients receiving hormonal maintenance therapy (primarily letrozole or tamoxifen) was 65 months, compared to 26 months for patients under observation only (*p* < 0.001) [82].

This study was followed up by Fader and colleagues, who also retrospectively explored the use of adjuvant hormonal therapy after surgery without the use of chemotherapy, with promising results [83]. A phase III trial (NRG GY 019) is now ongoing (initiated in 2019) and is comparing the treatment regimen of Paclitaxel/Carboplatin + Letrozole versus that of Letrozole alone in stage II–IV LGSOC [80].

### 5.5. Endometrioid Ovarian Carcinoma

Many endometrioid ovarian carcinomas have high expression of both ER and PR, and these patients generally demonstrate favorable survival outcomes [30,84]. It has been proposed that patients with stage II disease who have high PR status and would typically undergo systemic chemotherapy should be considered for possible endocrine therapy [84].

However, to date, there is still very limited information on the sensitivity of endometrioid ovarian carcinomas to hormonal therapies. Within the high grade ovarian carcinoma study at the Royal Marsden Hospital described above, five patients with high grade endometrioid ovarian carcinoma were treated with endocrine therapy, and encouragingly, three demonstrated a partial response while the other two had stable disease [77]. In the letrozole study reported by Bowman and colleagues, 4 of 11 endometrioid ovarian cancer patients had stable disease compared to 4 of 43 serous carcinoma patients [85].

A case report of two patients with advanced endometrioid ovarian carcinoma treated with letrozole reported one patient undergoing 30 months of remission before relapse, while the other patient was disease-free after 30 months of treatment [86]. These limited patient numbers indicate that further studies are warranted in this subgroup of patients.

### 5.6. Granulosa Cell Tumors

Estrogen-targeting therapies have shown marked promise in the treatment of GCTs [87]. In early studies prior to the use of aromatase inhibitors, tamoxifen produced a response rate (8%) and stable disease rate (32%) comparable to those obtained with epithelial ovarian cancer subtypes [87]. However, it is the aromatase inhibitors that are of great interest here. As mentioned above, one of the consequences of the FOXL2 mutation is an increase in aromatase activity in malignant granulosa cells, and this, leads through an undefined mechanism, to sensitivity to aromatase inhibitors [87]. In a review summarizing the use of the aromatase inhibitors as single agents, 25 cases with known outcomes are described. The response rate for these to the aromatase inhibitors was 48% (12 of 25), and the clinical benefit rate was 76% (19 of 25) [87]. A previous analysis had identified nine out of nine responders to aromatase inhibitors [88]. Although these patient numbers are limited, they clearly support the use of aromatase inhibitors as a potential alternative to chemotherapy.

## 6. Use of Estrogen-Regulated Predictive Biomarkers

Since only a percentage of ovarian carcinomas are responsive to anti-estrogen therapies, an ongoing challenge has been to identify and characterize those patients who will benefit from these therapies [73,76,89,90]. The search for predictive biomarkers to aid the selection of these patients is under investigation. Foremost among these biomarkers has been ERα itself, together with estrogen-regulated proteins, which are indicative of estrogen-regulated action. A number of these biomarkers have been assessed in clinical samples from trials evaluating the activity of letrozole [85,91,92,93,94], fulvestrant [95] or anastrazole in ovarian cancer patients [75]. Studies that have shown statistically significant associations are summarized in Table 1.

The most promising marker to date has been ERα expression itself. Preclinical data suggested that epithelial ovarian cancer cells with moderate to high ERα expression were growth-regulated by 17β-estradiol and were responsive to anti-estrogens [32,33,34,35]. This has recently been extended to include ovarian cancer explants [70]. In the first clinical phase II trial using letrozole in epithelial ovarian cancer, it was observed that the probability of obtaining clinical benefit (either a response or stable disease) was associated with an increased level of ERα expression [85]. This led to a follow-on Phase II study wherein only patients with a moderate–high ERα expression level were treated and a higher clinical benefit rate was obtained [93]. A high ERα histoscore was also associated with an increased treatment-free interval in a recent study reported by Stanley et al. [78], with some evidence of a dose–response effect regarding the extent of ERα positivity. Similarly, in the single Phase II trial of fulvestrant [94], the retrospective evaluation of ERα status demonstrated that clinical benefit was associated with higher-ERα-expressing cancers [95].

However, a number of trials have been unable to associate clinical response with ERα expression [75,77,96,97]. There are several potential reasons for the discrepancy in findings between studies. Firstly, in studies where an association between the clinical response and ERα expression was identified, a high level of expression of ERα, rather than simply any degree of ERα positivity, was key. In the studies that identified positive associations with the clinical response, semi-quantitative immunoscoring methods were used for assessing the percentage cell positivity and intensity of staining. This more detailed granular measurement of the level of ERα provides a more precise assessment of the level of receptor, which appears to be critical to the probability of a response. Secondly, since the percentage of responders is low, the analysis of the combination of stable disease with response to provide a measure of clinical benefit helped to provide sufficient statistical power to allow associations to be more readily identified. Furthermore, a recent study demonstrated that some patients display a delayed disease stabilization phenotype, where they initially progress but then stabilize, which appears to be clinically meaningful in terms of patient benefit [78]. Future trials might consider this possibility in their protocol design to allow this benefit to be seen, as most trial protocols would take patients off treatment early in progression [78]. Thirdly, the clinical trials have generally encompassed all histological groups, and since some subtypes (e.g., HGSOC, LGSOC and endometrioid carcinoma) are more responsive to endocrine therapy than others (e.g., mucinous or clear cell carcinomas), the relative proportions of different subtypes within a trial cohort treated will likely influence the final outcome.

In breast cancer, activating mutations in ERα (mutESR1) frequently contribute to therapeutic resistance, especially to aromatase inhibitors. Although rare in untreated ovarian cancers—e.g., mutation rates of 3.5% in endometrioid and 0.3% in serous ovarian cancers [98]—their appearance after treatment may have important therapeutic consequences. In a case report of a patient with LGSOC who had a sustained response to anastrazole therapy for 5 years, the patient developed an isolated recurrent lesion that was found to contain an *ESR1*-activating mutation (Y537S) [99]. Further LGSOCs and some HGSOCs that have developed mutESR1 after treatment with aromatase inhibitors have now been identified, and these include the activating mutations of Y537S, Y537N and D538G [98] (Figure 1). Since long-term therapy with aromatase inhibitors (AIs) may increase the frequency of these mutations, patients may need to be screened for their development. The presence of an activating mutation does not preclude the use of endocrine therapy; however, a switch to tamoxifen or fulvestrant treatment may be appropriate [98].

Since ERα expression alone is insufficient to identify the presence of estrogen regulation within ovarian cancers, downstream estrogen-regulated markers have been evaluated, as they may help inform which tumors are under estrogen growth control. Gene expression analysis has identified estrogen-regulated genes in ERα-positive ovarian carcinoma cells [38,70,91,92]. As for breast cancer, the progesterone receptor has been shown to be estrogen-regulated in ERα-positive ovarian carcinoma cells [33,100], and in the Phase II letrozole trial reported by Bowman et al. [85], higher progesterone receptor expression was associated with stable disease.

Among other estrogen-regulated markers, the IGFBPs appear to be particularly interesting [91]. Estrogen down-regulates IGFBP3 and IGFBP5 and up-regulates IGFBP4 in ERα-positive ovarian carcinoma cells. The letrozole response is similarly associated with these modified levels of expression in ERα-positive ovarian carcinomas [91], and this has been confirmed for both IGFBP3 [70] and IGFBP5 in subsequent trials [93]. Further estrogen-regulated markers that have been associated with the letrozole response within the trial setting are listed in Table 1. Increased expression of TFF1 (pS2) was associated with letrozole benefit [93], and this is known to be strongly regulated by estrogen in breast cancer. TFF1 expression has also been associated with a response to fulvestrant [95]. The other estrogen-regulated markers identified include TFF3, BIGH3, TRAP1, VIM, TOP2A, PLAU, UBE2C and CYP191A1, and these have all been shown to be differentially expressed in ovarian carcinomas, indicating clinical benefit from letrozole [92]. The expression of EGF receptor and HER2 is also linked to response, and these receptors are well known to interact with estrogen signaling in breast cancer [85,93]. Further prospective studies are now required to confirm and validate these potential biomarkers.

## 7. Conclusions

Estrogen signaling is functionally significant in the major histotypes of ovarian cancer, including HGSOC, LGSOC, endometrioid ovarian carcinoma and adult type GCTs. ERα is the dominant form of ER, promoting growth and migration, while wild-type ERβ mainly functions in growth inhibition. However, this is dependent on the nature of the ERβ isoforms present. Information on the role and impact of GPER1 is still limited, with evidence for both tumor-promoting and tumor-suppressing roles, and its significance relative to ERα and ERβ is unknown.

At present, anti-estrogen treatments such as the aromatase inhibitors and tamoxifen represent viable options for ER-positive patients (and are well tolerated and inexpensive) but are currently under-utilized as they are not the standard of care. With accumulating evidence that a percentage of patients within the above-mentioned subgroups could benefit from these treatments, it will be important to identify these patients with greater confidence if this strategy is to become more effective. As for breast cancer, the increased expression of ERα currently represents the best predictive molecular indicator of response; however, further biomarker studies are warranted to identify these estrogen-responsive cancers more precisely.

## Figures and Tables

**Figure 1 cancers-12-01647-f001:**
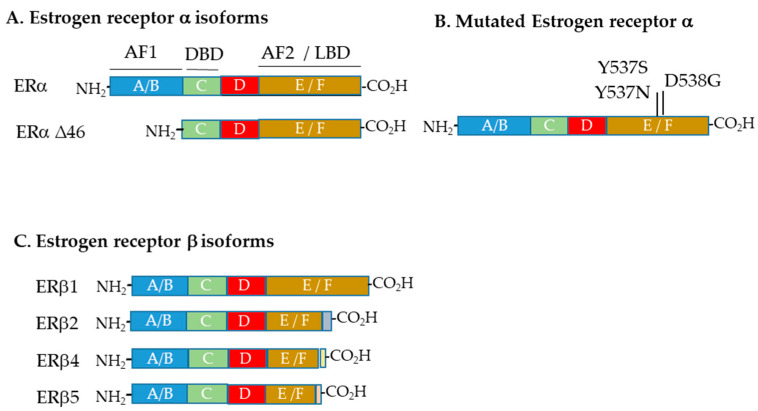
Estrogen receptor isoforms and mutants found within ovarian cancers. (**A**) Estrogen receptor alpha isoforms identified in ovarian cancers. (**B**) Mutated versions of estrogen receptor alpha found in ovarian cancers. Mutations at positions 537 and 538 have been observed. (**C**) Estrogen receptor beta isoforms identified in ovarian cancers. AF1 = Activation function 1; DBD = DNA binding domain; AF2 = Activation function 2; LBD = Ligand binding domain.

**Table 1 cancers-12-01647-t001:** Clinical studies of anti-estrogen therapies in ovarian cancer that have linked biomarker expression to clinical response.

Endocrine Agent	Clinical Study	Biomarkers Associated with Response	Biomarker Study
Letrozole	Bowman et al. [85]	ERα, PGR, EGFR, HER2, IGFBP3, IGFBP4, IGFBP5, TFF1, TFF3, BIGH3, TRAP1, VIM, TOP2A, PLAU, UBE2, CYP19A1	[85,91,92]
Letrozole	Smyth et al. [93]	ERα, HER2, IGFBP5, TFF1, VIM	[93]
Letrozole/Tamoxifen ^1^	Stanley et al. [78]	ERα	[78]
Tamoxifen/AIs ^2^	Andersen et al. [70]	ERα, IGFBP3	[70]
Fulvestrant	Argenta et al. [94]	ERα, TFF1, VIM	[95]

^1^ Composite analysis of letrozole (*n* = 207) and tamoxifen (*n* = 50) patients. ^2^ Composite analysis of tamoxifen (*n* = 59) and aromatase inhibitor (*n* = 18) patients. The aromatase inhibitors (AIs) used in this latter study were not reported.

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
