# Peer review of "Estrogen Signaling and Its Potential as a Target for Therapy in Ovarian Cancer"

_cancers, 2020, doi:10.3390/cancers12061647_

Round 1

Reviewer 1 Report

This review manuscript by Langdom et al., summarizes the recent advances in preclinical studies on the role of ERα, ERβ and GPER1 in ovarian cancer. Overall, the review article is relatively well written and easy to read. The authors treated the recent literatures on the topic carefully without any significant bias. Some commentary notes are also made appropriately. Literature synthesis is comprehensive, relatively well organized and interesting with enough details. Overall, the manuscript is suitable for publication with some minor changes as suggested below.

Some minor suggestions:

1) The title of the review needs to be modified, the review is not only dedicated to ERβ, but the authors are discussing also the role of other proteins ERα and GPER1. A more general title is necessary.

3) In figure 1 the size of writing should be standardized, some have smaller dimensions (e.g. E/F…)

4) The authors should add a list of abbreviations used.

5) Lines 196-198 “Use of SKOV3 and OV2008 cells which express both receptor subtypes were treated with subtype specific modulators”, review the sentence.

6) A careful spelling/typo check is needed.

Author Response

Comments and Suggestions for Authors

This review manuscript by Langdom et al., summarizes the recent advances in preclinical studies on the role of ERα, ERβ and GPER1 in ovarian cancer. Overall, the review article is relatively well written

and easy to read. The authors treated the recent literatures on the topic carefully without any significant bias. Some commentary notes are also made appropriately. Literature synthesis is comprehensive, relatively well organized and interesting with enough details. Overall, the manuscript is suitable for publication with some minor changes as suggested below.

  • Thank you for these comments

Some minor suggestions:

  • The title of the review needs to be modified, the review is not only dedicated to ERβ, but the authors are discussing also the role of other proteins ERα and GPER1. A more general title is necessary.

- We have modified the title by removing the word “receptor” to allow this to be more general.

2) In figure 1 the size of writing should be standardized, some have smaller dimensions (e.g. E/F…)

- We have corrected this

3) The authors should add a list of abbreviations used.

- We have added a list of abbreviations as suggested but were unsure if the format of the journal permitted this? If not, then the list can be deleted.

4) Lines 196-198 “Use of SKOV3 and OV2008 cells which express both receptor subtypes were treated with subtype specific modulators”, review the sentence.

We have modified this to ““The SKOV3 and OV2008 ovarian cancer cell lines express both receptor subtypes and these were treated with subtype specific modulators”

5) A careful spelling/typo check is needed.

We have checked through again.

Reviewer 2 Report

The present manuscript is a review article which reports the current knowledge about estrogen receptors and several aspects associated with ovarian cancer therapy. This review is informative and clear although the direct implication of estrogen receptors in ovarian cancer therapy is not clearly demonstrated.

However, some points described below should be included in order to improve this review:

  1. The impact of estrogen receptors on ovarian cancer is not clear especially in the case of ovarian cancer therapy. Authors should include not only clinical data but also evidence supporting the idea that estrogen receptors may be a target for therapy in ovarian cancer. Potential mechanisms mediated by estrogen receptors in the therapy of ovarian cancer should be presented. The point of view of authors in this aspect should be discussed helping the reader.

  1. Subsection 3.2 “ERβ expression in ovarian carcinomas” is too synthetic please rearrange.

  1. Figure 1. ERα36 is omitted. The type of mutation should be included.

  1. The text includes some particularly awkward sentences that should be rewritten to be clearer.
    1. Line 39-40: “Estrogen signaling is mediated by several estrogen receptors isoforms.. Estrogen receptor α (ERα, ESR1) was first identified…”
    2. Line 141-141: “Moreover, silencing of ERα through siRNA reduction removed the estrogen growth stimulation [37].”
    3. Line: 238-239: “In GCTs, ERβ is expressed more strongly that ERα and ERβ and has been proposed to be the dominant receptor [65].

Author Response

Comments and Suggestions for Authors

The present manuscript is a review article which reports the current knowledge about estrogen receptors and several aspects associated with ovarian cancer therapy. This review is informative and clear although the direct implication of estrogen receptors in ovarian cancer therapy is not clearly demonstrated.

However, some points described below should be included in order to improve this review:

The impact of estrogen receptors on ovarian cancer is not clear especially in the case of ovarian cancer therapy. Authors should include not only clinical data but also evidence supporting the idea that estrogen receptors may be a target for therapy in ovarian cancer. Potential mechanisms mediated by estrogen receptors in the therapy of ovarian cancer should be presented. The point of view of authors in this aspect should be discussed helping the reader.

  • We have added a new section (Section 5.1) to cover this

Subsection 3.2 “ERβ expression in ovarian carcinomas” is too synthetic please rearrange.

  • We have rearranged this section.

Figure 1. ERα36 is omitted. The type of mutation should be included.

In Figure 1, we wanted to just show the forms of ER-alpha and ER-beta that had been identified in ovarian cancer cells. To the best of our knowledge, ERa36 has not been observed – we have added a phrase in the text to emphasise this.

The text includes some particularly awkward sentences that should be rewritten to be clearer.

Line 39-40: “Estrogen signaling is mediated by several estrogen receptors isoforms.. Estrogen receptor α (ERα, ESR1) was first identified…”

We have modified as suggested by removing the words “ which possess different functionality“

Line 141-141: “Moreover, silencing of ERα through siRNA reduction removed the estrogen growth stimulation [37].”

We have changed to “Moreover, silencing of ERα through siRNA reduction neutralized the growth stimulation produced by added estrogen [37].

Line: 238-239: “In GCTs, ERβ is expressed more strongly that ERα and ERβ and has been proposed to be the dominant receptor”

We have changed to “In GCTs, ERβ is expressed more strongly than ERα and has led to the suggestion that ERβ is the dominant receptor”

Round 2

Reviewer 2 Report

The authors included all suggestions in the manuscript.